# Predicting Dam Flood Discharge Induced Ground Vibration with Modified Frequency Response Function

**Yan Zhang** [1,*] **, Jijian Lian** [2,3] **, Songhui Li** [1] **, Yanbing Zhao** [4] **, Guoxin Zhang** [1] **and Yi Liu** [1]

1   State Key Laboratory of Simulation and Regulation of Water Cycle in River Basin, China Institute of Water Resources and Hydropower Research, Beijing 100038, China; lish@iwhr.com (S.L.); zhanggx@iwhr.com (G.Z.); liuyi@iwhr.com (Y.L.)
2   State Key Laboratory of Hydraulic Engineering Simulation and Safety, Tianjin University, Tianjin 300072, China; fj_np@126.com or jjlian@tju.edu.cn
3   School of Water Conservancy and Hydropower, Hebei University of Engineering, Handan 056038, China
4   College of Water Resources Science and Engineering, Taiyuan University of Technology, Taiyuan 030024, China; zhaoyanbing@tyut.edu.cn
*   Correspondence: zcy881120@126.com or zhangyan@iwhr.com

**Abstract:** Ground vibrations induced by large flood discharge from a dam can damage surrounding buildings and impact the quality of life of local residents. If ground vibrations could be predicted during flood discharge, the ground vibration intensity could be mitigated by controlling or tuning the discharge conditions by, for example, changing the flow rate, changing the opening method of the orifice, and changing the upstream or downstream water level, thereby effectively preventing damage. This study proposes a prediction method with a modified frequency response function (FRF) and applies it to the in situ measured data of Xiangjiaba Dam. A multiple averaged power spectrum FRF (MP-FRF) is derived by analyzing four major factors when the FRF is used: noise, system nonlinearity, spectral leakages, and signal latency. The effects of the two types of vibration source as input are quantified. The impact of noise on the predicted amplitude is corrected based on the characteristics of the measured signal. The proposed method involves four steps: signal denoising, MP-FRF estimation, vibration prediction, and noise correction. The results show that when the vibration source and ground vibrations are broadband signals and two or more bands with relative high energies, the frequency distribution of ground vibration can be predicted with MP-FRF by filtering both the input and output. The amplitude prediction loss caused by filtering can be corrected by adding a constructed white noise signal to the prediction result. Compared with using the signal at multiple vibration sources after superimposed as input, using the main source as input improves the accuracy of the predicted frequency distribution. The proposed method can predict the dominant frequency and the frequency bands with relative high energies of the ground vibration downstream of Xiangjiaba Dam. The predicted amplitude error is 9.26%.

**Keywords:** ground vibration; flood discharge; vibration prediction; frequency response function; vibration source

## 1. Introduction

The construction of high dams with large reservoirs has increased globally owing to the increased utilization of water resources and demand for hydropower stations [1]. The water energy of the flood discharge from a high dam produces strong fluctuating flow loads. These loads can induce strong vibrations in discharge structures, and these vibrations are, in turn, transmitted to surrounding areas through the foundations of the dam. The effect of such vibrations is magnified under special geological conditions such as soft soil conditions [2,3]. Such vibrations can cause ground liquefaction, uneven foundation settlement, cracking of building walls, and harm to the health of residents [4–6]. Therefore, ground vibrations in areas near high dams must be investigated, monitored, and controlled.

Predicting the vibration induced by flood discharge before it transfers to the surrounding ground, and assessing its impact is an effective way to avoid or mitigate vibration-induced risks. Meanwhile, we hope to predict the vibration of the surrounding ground through the known vibration source, because it is more difficult to obtain the vibration data from the surrounding ground compared with that of the vibration source.

Many studies have predicted the ground vibrations induced by urban traffic [7–9]. However, only a few have predicted the ground vibrations induced by flood discharge [10]. In fact, ground vibrations induced by flood discharge can be analyzed as a structural vibration system with input, structure, and output in a manner analogous to the ground vibrations induced by urban traffic. In this case, the vibration system is described as follows: input (discharge excitation)—structure (discharge structure-dam-ground)—output (ground vibration). Therefore, the ground vibrations can be predicted by calculating the frequency response function (FRF) between the input and the output [11,12]. The FRF is a nonparametric estimation method that can reflect the transfer ability of a vibration system [13]; it is generally used to analyze the structural dynamic characteristics, for example, the structural modal parameter estimation [14,15] or the structural damage detection [16,17]. Owing to the characteristics of the FRF, researchers have applied the FRF for predicting the ground vibrations induced by urban traffic [18,19]. First, the FRF can be used to estimate both the intensity and the frequency distribution of ground vibrations [20,21]. In addition to the amplitude, it is more important to accurately predict the vibration frequency because potential resonance with buildings can be avoided by changing the vibration frequency. Second, the FRF can be used for real-time predictions because it remains constant when the input and output are fixed. Then, the unknown output can be estimated by the real-time obtained input and the calculated FRF. However, there are three major differences between the vibrations induced by flood discharge and those induced by urban traffic: (1) Flood discharge excitation is a random excitation with a broadband frequency, and varies with the discharge conditions. I addition, it is more susceptible to environmental factors such as noise and ambient vibration [2]. (2) The vibration transmission problem caused by flood discharge involves more influencing factors such as the coupling between the fluid and the solid. (3) The flood discharge excitation is the superposition of the vibrations generated when the flood flows through different parts of the discharge structures, for example, orifices, guide walls, and stilling pool base slabs [2], and these vibrations are interrelated and interactive in both time and space. Therefore, multiple inputs should be considered for flood discharge vibrations, whereas only a single input needs to be considered for the most urban traffic vibrations. These factors make it difficult to apply the FRF to predict the ground vibrations induced by flood discharge.

In this study, we propose a prediction method with a modified FRF and apply it to ground vibrations induced by flood discharge in the downstream area of Xiangjiaba Dam, a large-scale high dam located on the Jinsha River, Yunnan, China. A multiple averaged power spectrum FRF (MP-FRF) is derived by analyzing four major factors when FRF is used: noise, system nonlinearity, spectral leakages, and signal latency. The effects of two types of vibration source as input are quantified, e.g., the superimposed multiple vibration sources and the main vibration source. The impact of noise on the predicted amplitude is corrected based on the characteristics of the measured signal. The proposed method involves four steps: signal denoising, MP-FRF estimation, vibration prediction, and noise correction.

## 2. Background

### 2.1. Xiangjiaba Dam and Ground Vibrations Induced by Flood Discharge

Significant ground vibrations and some structural damage have been reported in Shuifu town near the Xiangjiaba hydropower station, the last cascade hydropower station on the Jinsha River, Yunnan, China. Xiangjiaba Dam is a high concrete gravity dam with a height, length, and crest elevation of 162.00 m, 909.26 m, and 384.00 m, respectively. Its normal storage water level is 380.00 m. The dam has 12 surface orifices and 10 midlevel

orifices; these are divided into two symmetrical energy dissipation zones by a middle guide wall. This type of layout of orifices is a typical one in high dams with energy dissipation zones for a hydraulic jump [10].

Figure 1 shows a map indicating the location of the dam and the displacement measurement points, including those in Shuifu town. Owing to a large number of measurement points in the dam area, only those at the orifice, on the foundation of the dam, and on the stilling pool base slab (SPBS) area are marked as T1, T2, and T3, respectively. The blue columns represent the relative RMS of the vertical displacement. Table 1 shows the RMS of the vertical displacement of measurement points. For comparison, the allowable vibration level for strict working area is converted to 0.005 μm in the standard ISO 2631-2: 2003 "Mechanical vibration and shock—Evaluation of human exposure to whole—body vibration—Part 2: Vibration in buildings (1 Hz to 80 Hz)". The minimum and maximum distances between the dam and the downstream area are 0.5 km and 3.0 km, respectively.

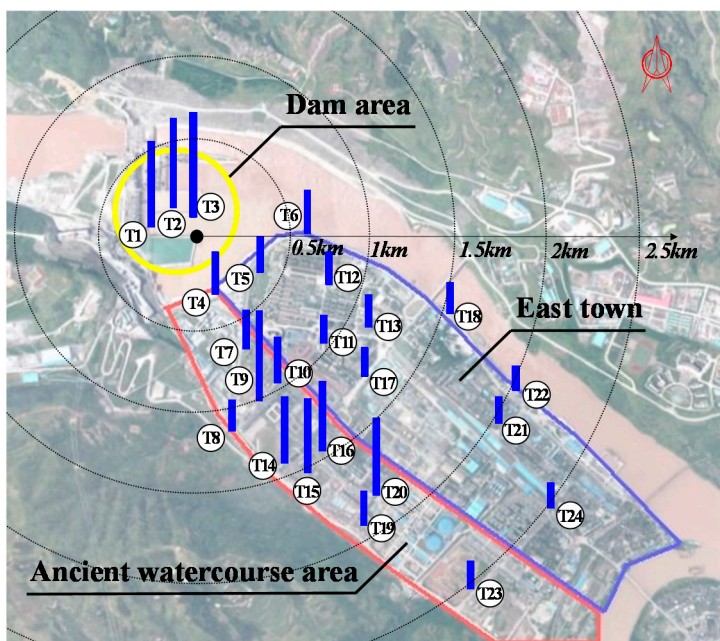

**Figure 1.** Map of Xiangjiaba Dam and Shuifu town and layout of the measurement points.

**Table 1.** The RMS of the vertical displacement of measurement points.

| Measurement Point | RMS of the Vertical Displacement (μm) | Measurement Point | RMS of the Vertical Displacement (μm) | Measurement Point | RMS of the Vertical Displacement (μm) |
|---|---|---|---|---|---|
| T1 | 1.33 | T9 | 1.24 | T17 | 0.34 |
| T2 | 1.48 | T10 | 0.53 | T18 | 0.37 |
| T3 | 1.81 | T11 | 0.33 | T19 | 0.40 |
| T4 | 0.50 | T12 | 0.38 | T20 | 0.89 |
| T5 | 0.40 | T13 | 0.38 | T21 | 0.31 |
| T6 | 0.50 | T14 | 0.76 | T22 | 0.29 |
| T7 | 0.46 | T15 | 0.85 | T23 | 0.33 |
| T8 | 0.36 | T16 | 0.81 | T24 | 0.29 |

Local residents have reported feeling vibrations in the mountainside area downstream of the dam. Geological studies of this area revealed that its surface and foundation consist of sand layers and mudstones. An ancient watercourse with an average width of ~200 m is located on the right side of the mountain. The maximum thickness of the covering layer over this watercourse is 80 m, whereas that of the covering layer in the east town adjacent

to this watercourse is significantly smaller at 20 m. The vibration intensity in the dam area is found to be larger than that in the downstream town area, and the vibration intensity in the ancient watercourse area is generally larger than that in the east town area owing to the magnified effect of its soft soil foundation.

### 2.2. Sensors and the Signal Collection System

The sensor used is the DP low-frequency vibration displacement sensor. The sensor types are DPS-0.2-5-V-A (vertical direction) and DPS-0.2-5-H-A (horizontal direction). The sampling frequency range is 0.35–200 Hz, the test error is $\leq \pm 0.1\%$ F.S., the sensitivity is 5 mV/μm.

The vibration response is collected by the vibration displacement sensor installed on the surface of the hydraulic structures and surrounding ground. The sampling time is 120 s, the sampling frequency is 80 Hz. The sensor converts the vibration signal into the voltage signal, and the digital signal of the vibration displacement is obtained after the signal collection system auto processed. Figure 2 shows the sensor installation and the signal collection system.

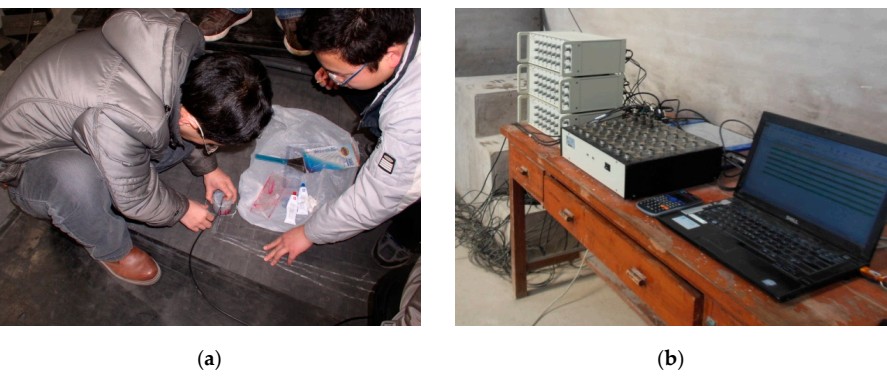

(**a**)            (**b**)

**Figure 2.** Sensor installation (**a**) and the signal collection system (**b**).

## 3. Prediction Methods with Modified FRF

### 3.1. Principles of FRF

The characteristics of the vibration transmission system are inherent [22]. This means that the system remains constant when the vibration source (input) and the vibration response (output) are selected. Figure 3 shows a ground vibration prediction process using the FRF when input and output don't contain noise.

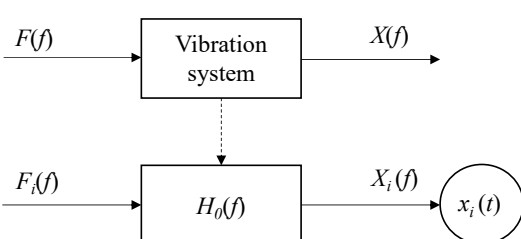

**Figure 3.** Prediction process when input and output don't contain noise.

$F(f)$ and $X(f)$ are the Fourier spectrum of the input and output signal, respectively. $H_0(f)$ is the FRF between $F(f)$ and $X(f)$. Therefore, $H_0(f)$ can be estimated using Equation (1) below. $H_0(f)$ is dimensionless if and only if both the input and the output are the same physical quantity; otherwise, it has dimensions.

$$H_0(f) = \frac{X(f)}{F(f)} \tag{1}$$

The new output Fourier spectrum $X_i(f)$ of the same measurement point can be obtained as

$$X_i(f) = H_0(f)F_i(f) \tag{2}$$

Other vibration information such as the time history curves $x_i(t)$ can be obtained by using an inverse Fourier transform [23–25].

### 3.2. The Influencing Factors and the Multiple Averaged Power Spectrum FRF

### 3.2.1. Influencing Factors

As the FRF is estimated using in situ measurement signals, its accuracy will inevitably be affected by interferences such as noise, system nonlinearity, spectral leakages, and signal latency [26,27].

FRF is usually divided into the unbiased estimation or the biased estimation type depending on whether noise is considered or not, respectively. Unbiased estimation should be adopted when both the input and the output contain noise and its effect cannot be ignored, as in the case of ground vibrations induced by flood discharge. Distinguishing between the linear and the nonlinear parts in the input and output signals is very important in calculating the FRF [28]. The coherence functions [29] can reflect the linear degree of the vibration transfer system at different frequencies. Spectral leakage is always caused by the time domain truncation of the signal. This is traditionally solved by windowing in the time domain, such as by using the Hanning window [30,31].

The effect of vibration signal latency on the long distance propagation cannot be ignored. The time difference between the two vibrations is influenced by the topography, the geology, the distance, etc. The correlation coefficient is generally used to describe the correlation degree between the two signals under different phases. Generally, the propagation speed of the vibration in the ground is between 1–6 km/s. In the case of the ground vibration happened near Xiangjiaba Dam, we calculate the correlation coefficient of different phase difference between the vibration source and ground measurement points at different distance. The time interval is 0.125 s, the maximum phase difference is 2.5 s. The variation of correlation coefficient with phase difference is shown in Figure 4, and the correlation coefficients of different measurement points in different phase differences are shown in Table 2. It can be seen that the correlation coefficient of each measurement point is between 0.2–0.6. The changing law of correlation coefficient of each measurement point with phase difference is not the same. The phase difference of the maximum correlation coefficient is between 0 s–1.5 s. Under the influence of topography and geology, the phase difference of the maximum correlation coefficient is different even at the same distance from the vibration source. Therefore, when we calculate the MP-FRF, we take the phase difference of the maximum correlation coefficient of each measurement point as the signal latency.

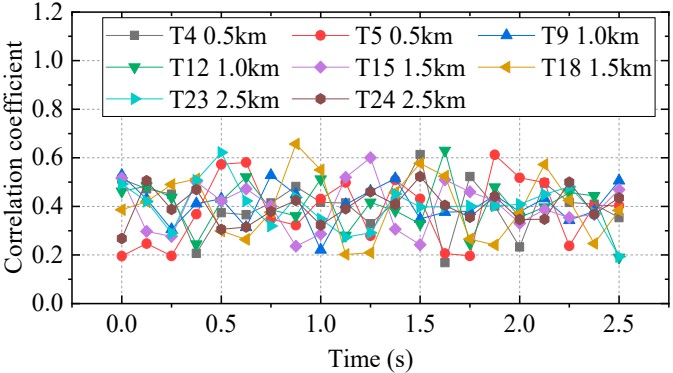

**Figure 4.** The variation of correlation coefficient with phase difference.

**Table 2.** The correlation coefficient of different measurement points in different phase differences.

| Measurement Point / Phase Difference (s) | T4 | T5 | T9 | T12 | T15 | T18 | T23 | T24 |
|---|---|---|---|---|---|---|---|---|
| 0 | 0.513 | 0.195 | 0.528 | 0.462 | 0.516 | 0.386 | 0.492 | 0.268 |
| 0.125 | 0.473 | 0.247 | 0.431 | 0.484 | 0.297 | 0.419 | 0.421 | 0.506 |
| 0.25 | 0.451 | 0.196 | 0.306 | 0.437 | 0.277 | 0.490 | 0.290 | 0.388 |
| 0.375 | 0.207 | 0.368 | 0.411 | 0.245 | 0.501 | 0.513 | 0.507 | 0.470 |
| 0.5 | 0.374 | 0.573 | 0.431 | 0.421 | 0.422 | 0.299 | 0.622 | 0.306 |
| 0.625 | 0.366 | 0.581 | 0.314 | 0.520 | 0.472 | 0.264 | 0.422 | 0.315 |
| 0.75 | 0.414 | 0.348 | 0.529 | 0.387 | 0.402 | 0.376 | 0.319 | 0.379 |
| 0.875 | 0.482 | 0.323 | 0.450 | 0.361 | 0.236 | 0.657 | 0.437 | 0.425 |
| 1 | 0.417 | 0.431 | 0.221 | 0.512 | 0.286 | 0.550 | 0.348 | 0.323 |
| 1.125 | 0.414 | 0.498 | 0.408 | 0.280 | 0.520 | 0.202 | 0.273 | 0.390 |
| 1.25 | 0.328 | 0.279 | 0.463 | 0.416 | 0.600 | 0.209 | 0.291 | 0.460 |
| 1.375 | 0.420 | 0.509 | 0.515 | 0.382 | 0.306 | 0.458 | 0.451 | 0.408 |
| 1.5 | 0.613 | 0.431 | 0.349 | 0.328 | 0.242 | 0.577 | 0.397 | 0.523 |
| 1.625 | 0.168 | 0.206 | 0.376 | 0.630 | 0.510 | 0.524 | 0.398 | 0.404 |
| 1.75 | 0.523 | 0.196 | 0.379 | 0.247 | 0.460 | 0.266 | 0.402 | 0.355 |
| 1.875 | 0.402 | 0.613 | 0.446 | 0.479 | 0.423 | 0.241 | 0.402 | 0.441 |
| 2 | 0.233 | 0.518 | 0.378 | 0.355 | 0.330 | 0.372 | 0.408 | 0.346 |
| 2.125 | 0.496 | 0.498 | 0.433 | 0.397 | 0.387 | 0.572 | 0.449 | 0.347 |
| 2.25 | 0.416 | 0.238 | 0.343 | 0.456 | 0.354 | 0.425 | 0.481 | 0.500 |
| 2.375 | 0.409 | 0.405 | 0.380 | 0.444 | 0.367 | 0.247 | 0.373 | 0.368 |
| 2.5 | 0.354 | 0.403 | 0.505 | 0.190 | 0.469 | 0.384 | 0.193 | 0.435 |

### 3.2.2. The Multiple Averaged Power Spectrum FRF

Figure 5 shows a prediction process when the input and output contain noises. The input $F(f)$ contains two components: The Fourier spectrum of the useful signal $Z(f)$ and the Fourier spectrum of background noise $M(f)$. The output $X(f)$ also contains two components: The Fourier spectrum of the useful signal $Y(f)$ and the Fourier spectrum of background noise $N(f)$.

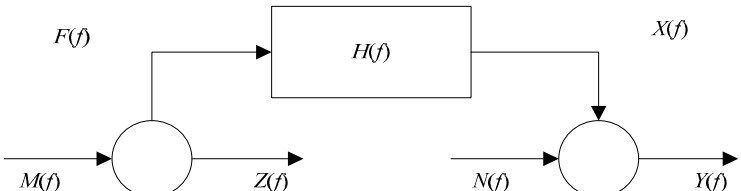

**Figure 5.** Prediction process when input and output contain noise.

Therefore, the FRF of the system can be expressed as

$$H(f) = \frac{Y(f)}{Z(f)} = \frac{X(f) - N(f)}{F(f) - M(f)} = H_0(f)\frac{1 - N(f)/X(f)}{1 - M(f)/F(f)} \tag{3}$$

Multiple tests were performed on the same vibration system by using the same excitation signal for the input of all tests. For the $i$-th measurement $(i = 1, 2, \ldots, n-1)$,

$$Y_i(f) - N_i(f) = H(f)[Z_i(f) - M_i(f)](i = 1, 2, \ldots, n-1) \tag{4}$$

where $F_i(f)$ and $X_i(f)$ are the $i$-th measurement of $F(f)$ and $X(f)$, respectively. $Z_i(f)$, $M_i(f)$, $Y_i(f)$ and $N_i(f)$ are the $i$-th measurement of $Z(f)$, $M(f)$, $Y(f)$ and $N(f)$, respectively. For the linear transfer system, $F_i(f) = F(f)$ and $X_i(f) = X(f)$ when the input is the same.

Multiplying both ends of the above equation by the conjugation of $Z_i(f)$, $Z_j^H(f)$, $(j = i + 1, 2, \ldots, n)$, and then dividing its expected value by half the sampling time $T$ gives

$$G_{ZY}(f) - G_{ZN}(f) = H(f)[G_{ZZ}(f) - G_{ZM}(f)] \tag{5}$$

where $G_{ZY}(f) = \frac{4}{T(n-1)n} \sum_{i=1}^{n-1} \sum_{j=i+1}^{n} Y_i(f) Z_j^{\mathrm{H}}(f)$ is the cross-power spectrum (CS) between $Z(f)$ and $Y(f)$, $G_{ZN}(f) = \frac{4}{T(n-1)n} \sum_{i=1}^{n-1} \sum_{j=i+1}^{n} N_i(f) Z_j^{\mathrm{H}}(f)$ is the CS between $Z(f)$ and $N(f)$, $G_{ZZ}(f) = \frac{4}{T(n-1)n} \sum_{i=1}^{n-1} \sum_{j=i+1}^{n} Z_i(f) Z_j^{\mathrm{H}}(f)$ is the auto-power (AS) spectrum of $Z(f)$, and $G_{ZM}(f) = \frac{4}{T(n-1)n} \sum_{i=1}^{n-1} \sum_{j=i+1}^{n} M_i(f) Z_j^{\mathrm{H}}(f)$ is the CS between $Z(f)$ and $M(f)$.

Owing to the randomness of noise, after a sufficient number of CS and AS transforms, the two noise signals cancel each other. Therefore, after multiple rounds of averaging, $G_{ZN}(f) = 0$, $G_{ZM}(f) = 0$, and Equation (5) can be changed to

$$G_{ZY}(f) = H(f) G_{ZZ}(f) \tag{6}$$

$H(f)$ can be obtained as

$$H(f) = G_{ZY}(f)/G_{ZZ}(f) \tag{7}$$

Equation (6) can also be expressed as

$$H(f) = \frac{G_{ZY}(f)}{G_{ZZ}(f)} = \frac{G_{FX}(f) + G_{MX}(f) + G_{FN}(f) + G_{MN}(f)}{G_{FF}(f) + G_{MF}(f) + G_{FM}(f) + G_{MM}(f)} \tag{8}$$

where $G_{FX}(f) = \frac{4}{T(n-1)n} \sum_{i=1}^{n-1} \sum_{j=i+1}^{n} X_i(f) F_j^{\mathrm{H}}(f)$ is the CS between $F(f)$ and $X(f)$, $G_{MX}(f) = \frac{4}{T(n-1)n} \sum_{i=1}^{n-1} \sum_{j=i+1}^{n} X_i(f) M_j^{\mathrm{H}}(f)$ is the CS between $M(f)$ and $X(f)$, $G_{FN}(f) = \frac{4}{T(n-1)n} \sum_{i=1}^{n-1} \sum_{j=i+1}^{n} N_i(f) F_j^{\mathrm{H}}(f)$ is the CS between $F(f)$ and $N(f)$, $G_{MN}(f) = \frac{4}{T(n-1)n} \sum_{i=1}^{n-1} \sum_{j=i+1}^{n} N_i(f) M_j^{\mathrm{H}}(f)$ is the CS between $M(f)$ and $N(f)$, $G_{FF}(f) = \frac{4}{T(n-1)n} \sum_{i=1}^{n-1} \sum_{j=i+1}^{n} F_i(f) F_j^{\mathrm{H}}(f)$ is the AS spectrum of $F(f)$, $G_{MF}(f) = \frac{4}{T(n-1)n} \sum_{i=1}^{n-1} \sum_{j=i+1}^{n} F_i(f) M_j^{\mathrm{H}}(f)$ is the CS between $M(f)$ and $F(f)$, $G_{FM}(f) = \frac{4}{T(n-1)n} \sum_{i=1}^{n-1} \sum_{j=i+1}^{n} M_i(f) F_j^{\mathrm{H}}(f)$ is the CS between $F(f)$ and $M(f)$, and $G_{MM}(f) = \frac{4}{T(n-1)n} \sum_{i=1}^{n-1} \sum_{j=i+1}^{n} M_i(f) M_j^{\mathrm{H}}(f)$ is the AS of $M(f)$.

Similarly, $G_{MX}(f) = 0$, $G_{FN}(f) = 0$, $G_{MN}(f) = 0$, $G_{MF}(f) = 0$, $G_{FM}(f) = 0$, and $G_{MM}(f) = 0$. Equation (8) can be rewritten as

$$H(f) = G_{FX}(f)/G_{FF}(f) \tag{9}$$

Equation (9) shows that the influence of noise has been completely eliminated; therefore, $H(f)$ is an unbiased estimation. The estimation accuracy depends not only on the CS between the input and the output and the AS of the input but also on multiple rounds of averaging. This requires several tests with the same input under similar conditions.

### 3.3. Input Comparison

Figure 6 shows the layout of the Xiangjiaba Dam and the possible vibration sources of downstream. The fluctuating loads at the orifices, SPBS, guide wall, tail-weir, and fall-sill are the main sources affecting ground vibrations. The vibration energy acting on the orifice is much higher than that at other positions under different discharge modes. Further, its contribution to ground vibrations exceeds the sum of the remaining vibration sources [2].

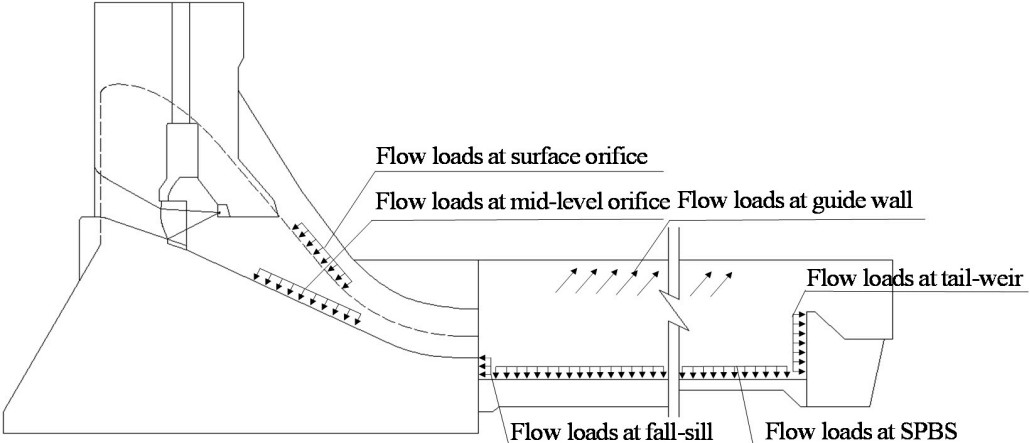

**Figure 6.** Location of different flow fluctuating loads.

### 3.3.1. Multiple Vibration Sources as an Input

The input should be a single signal when estimating the FRF. However, the vibration source includes 5 parts. Therefore, we try to use the signals at all vibration sources as an input because compared with the distance between each vibration source and the ground, the distance between the vibration sources is very small. In the present work, the simultaneous signals at 5 parts are superimposed in the time domain as a single input. The displacement of measurement point T9 with the largest vibration amplitude in Figure 1 is selected as the output. Further, 12 similar flood discharge conditions are averaged when estimating the MP-FRF. The similar flood discharge condition means the orifices opening mode and the orifices opening degree are the same, the flow rate and the upstream water level are close (the difference of flow rate is less than 50 m$^3$/s and the difference of upstream water level is less than 0.2 m) during flood discharge. $G_{FX}(f)$ is the averaged CS of the displacement of superimposed multiple sources and the displacement at T9, $G_{FF}(f)$ is averaged AS of the displacement of the superimposed multiple sources. $H(f)$ is the ratio of $G_{FX}(f)$ $G_{FF}(f)$. Figure 7 shows the time history curve and CS of $G_{FX}(f)$. Figure 7b indicates that the vibration frequency still shows a flow-load-induced vibration after the vibration source signals are superimposed.

Figure 8 are the MP-FRF and its coherence coefficients. In Figure 8b, all coherence coefficients lie in the range of 0–0.6 in the frequency band of 0–15.0 Hz and that the coherence coefficients in the frequency band of 20.0–40.0 Hz are reduced to 0–0.2. However, a few frequencies have large coherence coefficients; for example, the coherence coefficient at 21.0 Hz is ~0.4, possibly because of the influence of noise. For the MP-FRF (Figure 8a), the frequency band 1.0–6.5 Hz shows a relative high energy and the peak is at ~18 Hz. Large values are also seen at 5.5 Hz, 6.3 Hz, 22.0 Hz, and 29.5 Hz, indicating that ground vibrations are amplified at these frequencies.

Figure 9 shows the time history curves and Fourier spectra of the prediction result and the in-situ measurement of T9.

The amplitude of the prediction result is seen to be significantly larger than that of the in-situ measurement. The RMS of displacement of the prediction and in situ measurement are 0.122 μm and 0.047 μm, respectively. The predicted frequencies have many more peaks than the in-situ measurement one at both low and high frequencies. Specifically, the in-situ measurement has only one frequency band of 0–6.0 Hz with a relative high energy and the peak is at ~3 Hz, whereas the predicted signal has two frequency bands of 0–6.0 Hz and 8.0–12.0 Hz with relative high energies and the peak is at 1.7Hz. This may be because (1) non-source-induced vibrations are amplified when the vibration source signals are superimposed, and (2) white noise still exists at the input and output after multiple averaging. For example, a peak value at ~18 Hz is appeared in the MP-FRF (Figure 8). This causes a relative high vibration energy at ~18 Hz in the predicted Fourier spectra (Figure 9b), but this frequency is not in the in-situ measurement.

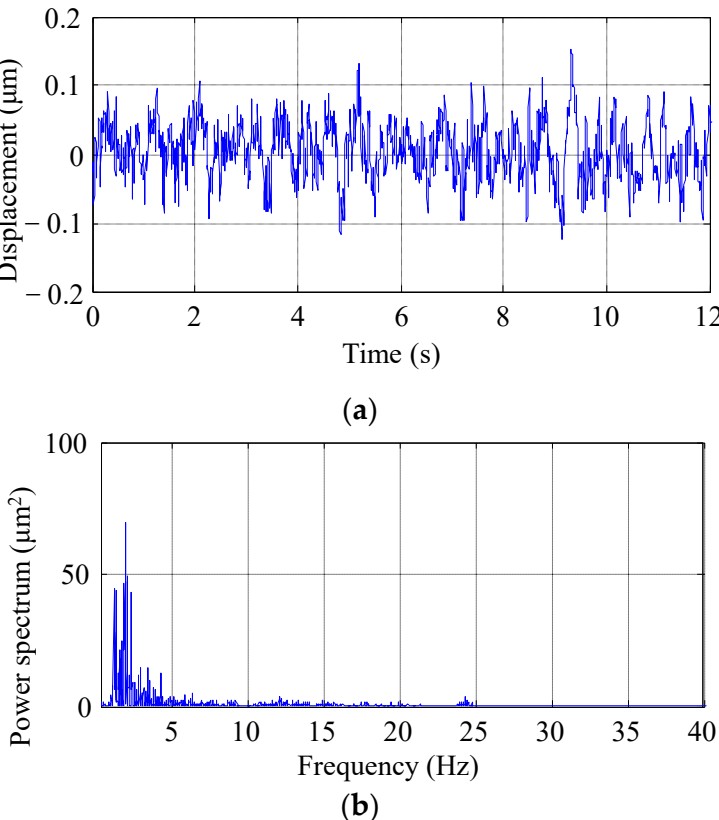

**Figure 7.** Superimposed signal of multiple vibration sources: (**a**) Time history curve; (**b**) CS.

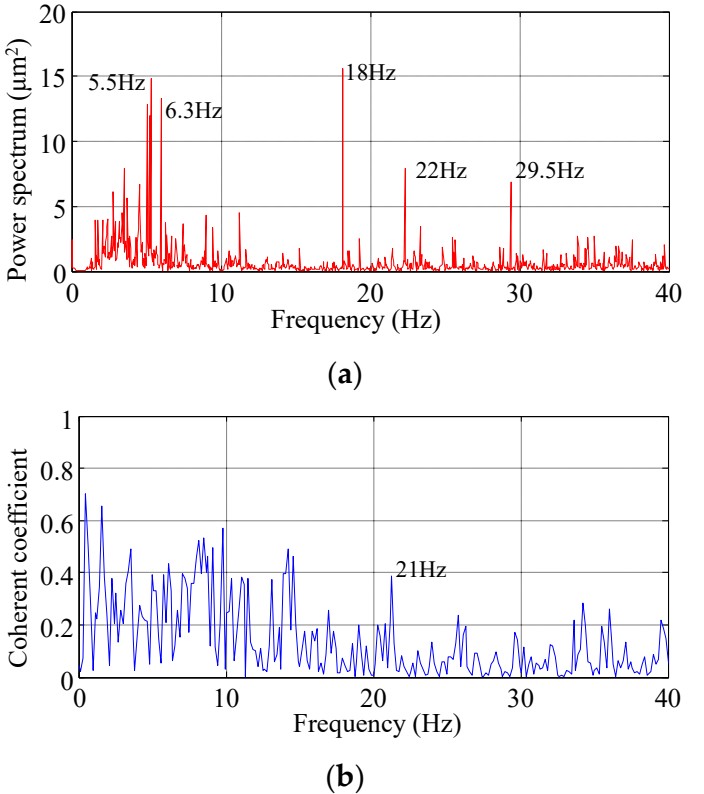

**Figure 8.** MP-FRF (**a**) and coherence coefficient (**b**) between the superimposed vibration source and T9.

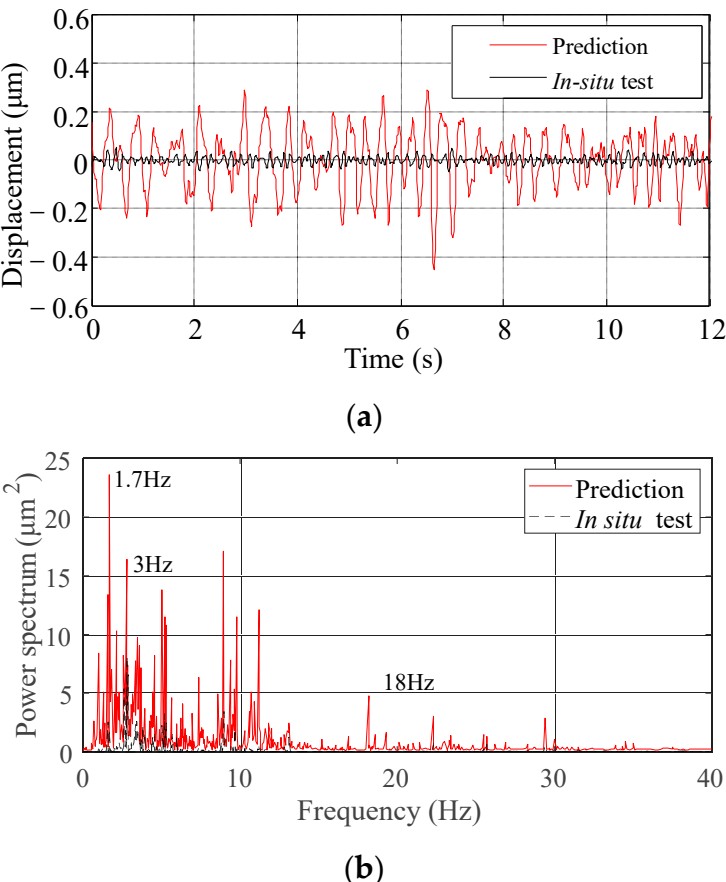

**Figure 9.** The signal of the prediction and the in situ measurement of T9 by using the superimposed vibration source as the input: (**a**) Time history curves; (**b**) and Fourier spectra.

3.3.2. Main Vibration Source as an Input

Another input mode is to use the main vibration source along. The vibration at the orifice is the main source of ground vibrations downstream of Xiangjiaba Dam. In this case, $G_{FX}(f)$ is the averaged CS of the displacement at the orifice and the displacement at T9, $G_{FF}(f)$ is averaged AS of the displacement at the orifice. $H(f)$ is the ratio of $G_{FX}(f)$ to $G_{FF}(f)$. Figure 10 shows the MP-FRF and its coherence coefficients.

Unlike the results obtained with the superimposed signals as the input, the coherence coefficient in the frequency band of 0–10.0 Hz increases to 0–0.8, indicating that the frequency of the band has less loss during the transfer process. In addition, the coherence coefficient in the frequency band of 10.0–40.0 Hz is generally less than 0.2. In terms of the MP-FRF, the frequency band 1.5–7.5 Hz with a relative high energy and the peak is at ~3 Hz. Large values also occur at 4.1 Hz, 7.2 Hz, and 18 Hz. Overall, there are some other large value peaks (larger than 2 $\mu m^2$) in the frequency band of 10.0–40.0 Hz. Figure 11 shows the time history curves and Fourier spectra of the prediction result and the in-situ measurement of T9.

The amplitude of the prediction result is also seen to be larger than that of the in-situ measurement. The RMS of displacement of the prediction and in situ measurement are 0.142 μm and 0.064 μm, respectively. We consider that the noise still has an impact on the prediction of vibration. In terms of the vibration frequency distribution, the frequency band 1.0–6.0 Hz with a relative high energy in the in-situ measurement is successfully predicted, and the main frequency of T9 of ~3 Hz is accurately predicted. However, a peak value at 4.1 Hz does not appear in the prediction.

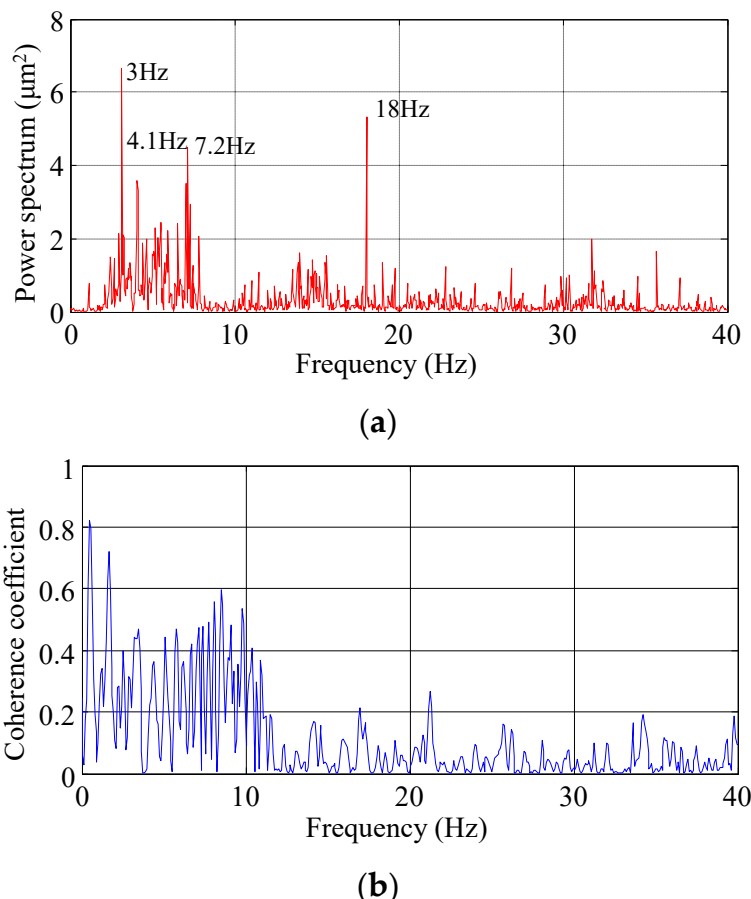

**Figure 10.** MP-FRF (**a**) and coherence coefficient (**b**) between the main source at orifice and T9.

Therefore, compared with the superimposed multiple sources as input, the accuracy of the predicted frequency is improved when the main source is used as the input. Further, when the input or output has two or more frequency bands with relative high energies, the prediction results are more easily affected by factors such as noise that make the prediction amplitude too large. Therefore, the input and output signals need to be preprocessed before using MP-FRF.

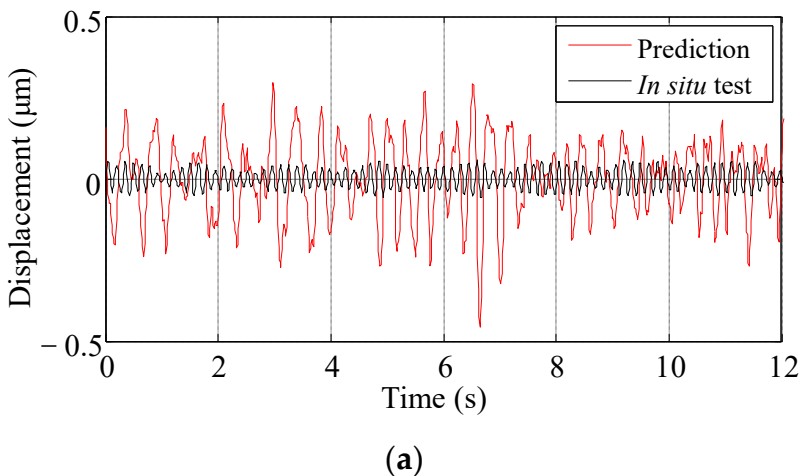

**Figure 11.** *Cont*.

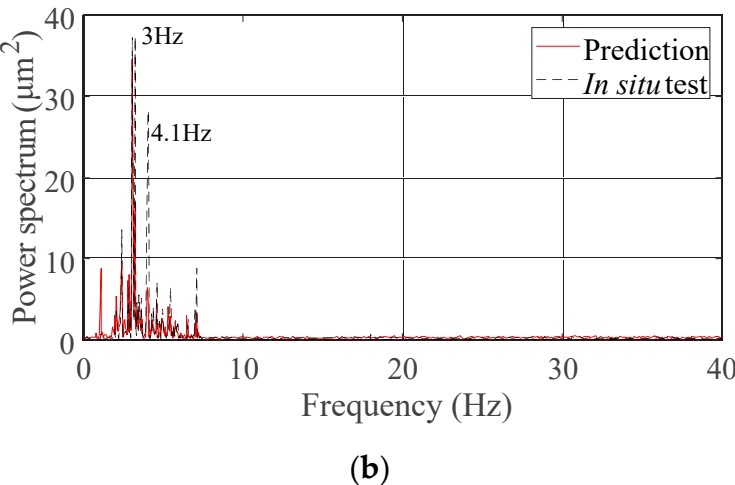

**(b)**

**Figure 11.** The signal of the prediction and the in situ measurement of T9 by using the main vibration sources as the input: (**a**) Time history curves; (**b**) and Fourier spectra.

### 3.4. Noise Correction

As the noises in the input and output are filtered at the same time when using MP-FRF, the predicted output is noise-free. Therefore, the effect of noise on the vibration amplitude should be reconsidered. A noise signal based on the signal-to-noise ratio (SNR) formula can be created to simulate real noise and then be added to the prediction signal. The SNR of the signal after noise reduction is

$$SNR = 20\ln\left(\frac{\sigma_y^2}{\sigma_{x-y}^2}\right) \tag{10}$$

where $\sigma_y^2$ is the variance of the time series after noise reduction, and $\sigma_{x-y}^2$ is the variance of the noise series after noise reduction. Therefore, the variance of the noise signal can be expressed as

$$\sigma_{x-y}^2 = \sigma_y^2 e^{-\frac{SNR}{20}} \tag{11}$$

### 3.5. Prediction Method with Modified FRF

Based on the above analysis, the ground vibration induced by flood discharge from the dam is predicted by using the following four steps:

1.  Signal denoising: A modified ensemble empirical mode decomposition (EEMD) and a wavelet threshold filtering method [32] are applied to filter the noise of the main source signals and the ground vibration signals.
2.  MP-FRF estimation: The CS of the input and output ($G_{FX}(f)$) and the averaged AS of the input ($G_{FF}(f)$) are calculated. The same is done for other similar discharge conditions. All CSs and ASs are averaged separately, the MP-FRF is estimated using Equation (9).
3.  Vibration prediction: The Fourier spectrum of the ground vibration can be obtained by multiplying the Fourier spectrum of the vibration source and the MP-FRF. The time history curve of the ground vibration can be obtained using an inverse Fourier transform.
4.  Noise correction: A noise signal calculated using Equation (11) is added to the predicted time history curve to correct the vibration energy loss due to filtering.

The detailed flow chart for prediction of ground vibration induced by flood discharge is shown in Figure 12.

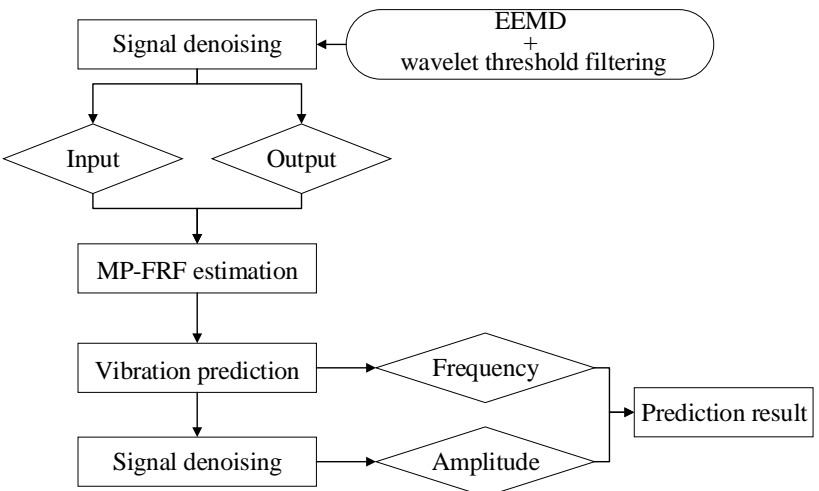

**Figure 12.** The flow chart for prediction of ground vibration induced by flood discharge.

## 4. Application to Ground Vibration Downstream of Xiangjiaba Dam
*Prediction Results*

The displacement at the orifice and T9 are filtered using the modified EEMD and wavelet threshold filtering method. Figure 13 shows their time-history curves before and after noise reduction. Figure 14 shows their Fourier spectra before and after noise reduction.

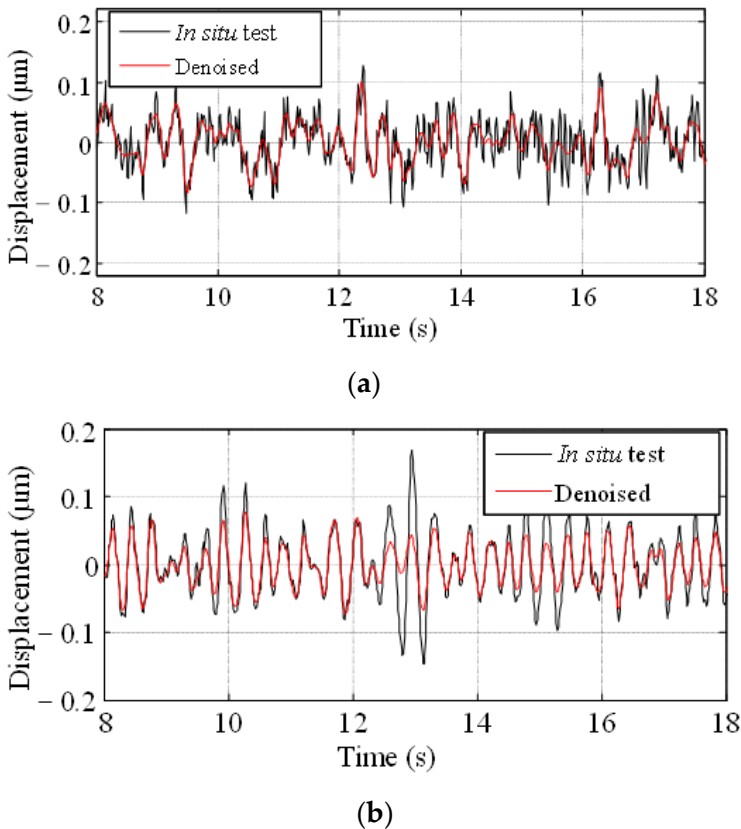

**Figure 13.** Time history curves of original signal and denoised signal at orifice (**a**) and T9 (**b**).

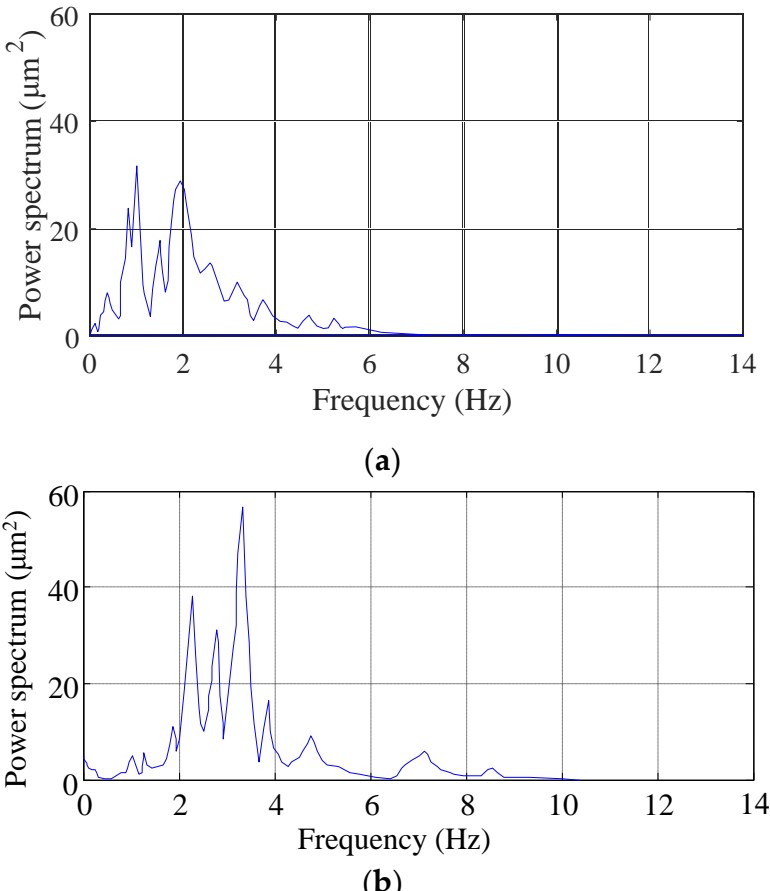

**Figure 14.** Fourier spectrum of denoised signal at orifice (**a**) and T9 (**b**).

The denoised input and output of the similar 12 flood discharge conditions mentioned in Section 3.3 are averaged. Figure 15 shows the MP-FRF and its coherence coefficients. The value of the coherence coefficient is seen to be between 0 and 0.7 in the frequency band of 0–10.0 Hz with a peak at 3.5 Hz. The coherence coefficient decreases gradually after 10.0 Hz. Overall, its value is less than 0.2. For the MP-FRF (Figure 15a), the distinct peak is at 3.5 Hz. The main transmission energy is concentrated in the frequency band of 1.0–4.0 Hz. A few vibrations with a relative high energy are at high frequencies like 16.5 Hz, 24.2 Hz, and 24.9 Hz.

Figure 16a shows the RMS of the displacements at T9 under different in situ measurements before and after filtering from small to large. Figure 16b shows the calculated SNR under the corresponding conditions. The SNR is seen to remain at ~15.90 dB, and it does not change significantly with the discharge conditions. According to this characteristic, the noise can be constructed based on Equation (11) and then be added to the predicted result, where the SNR is 15.90 dB.

Figure 17 shows a comparison of the prediction result and the in-situ measurement of the displacement at T9. In terms of the vibration amplitude (Figure 17a), the prediction result after adding noise shows good agreement with the in-situ ones. Both RMS of displacements are close, the prediction result is 0.142 μm and the in-situ measurement is 0.149 μm. In Figure 17b, two dominant frequencies at 1.8 Hz and 3.5 Hz are accurately predicted. The spectrum distribution in the frequency band of 0–10.0 Hz is similar to that of the in-situ measurements.

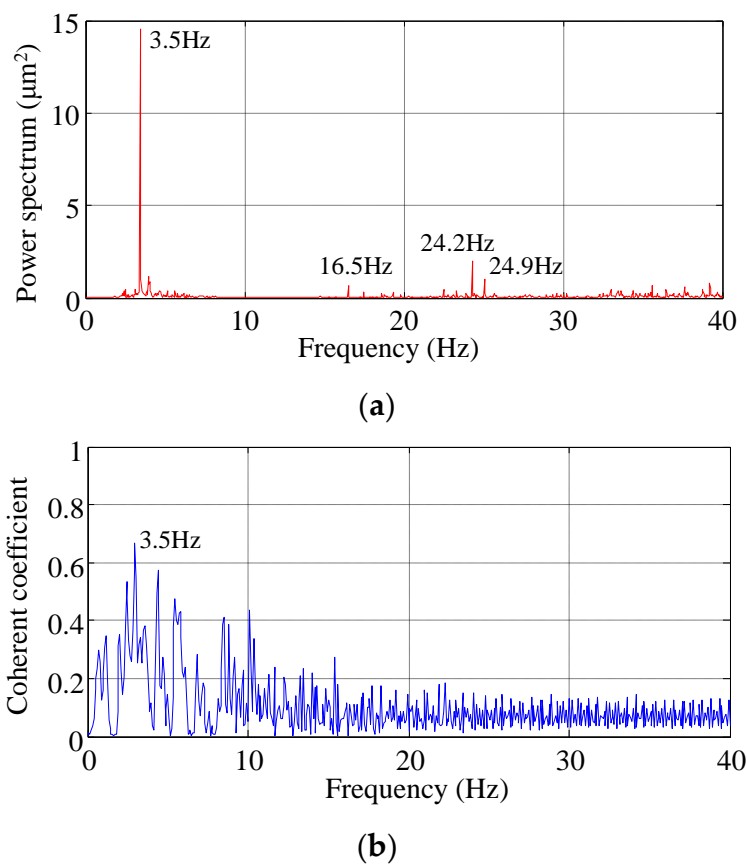

(**a**)

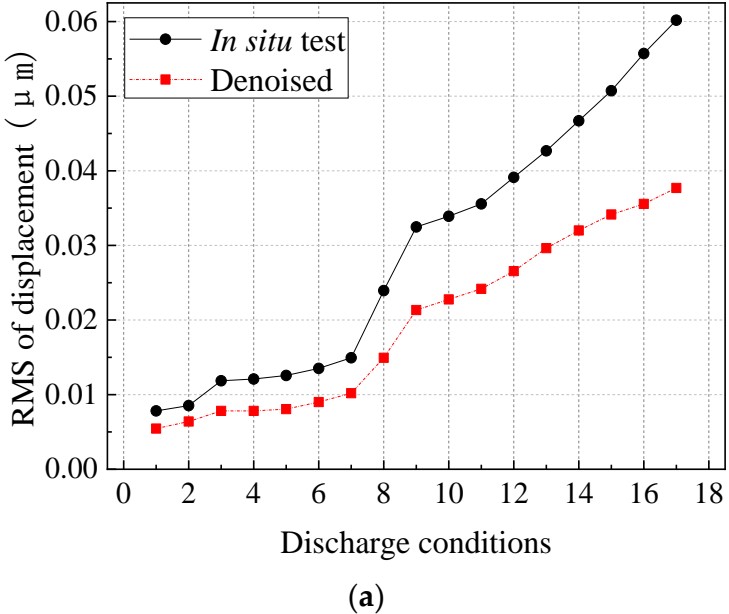

(**b**)

**Figure 15.** MP-FRF (**a**) and coherence coefficient (**b**) between the signal at orifice and T9.

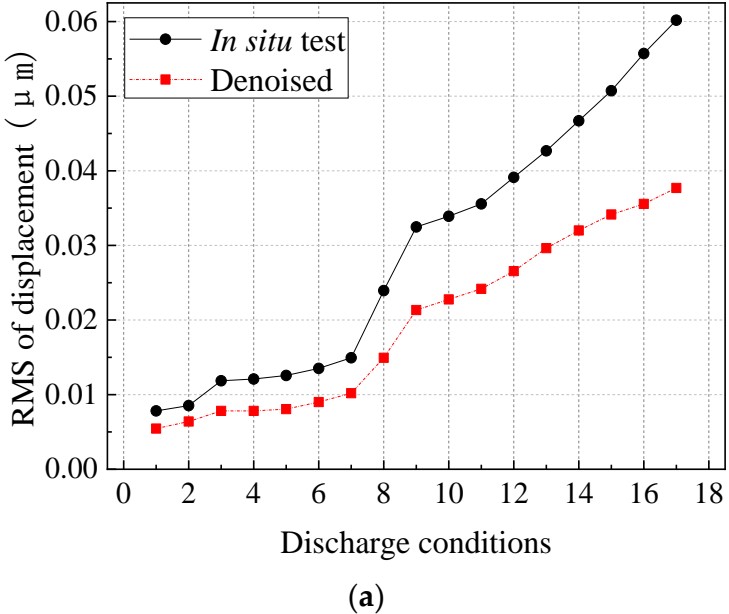

(**a**)

**Figure 16.** *Cont*.

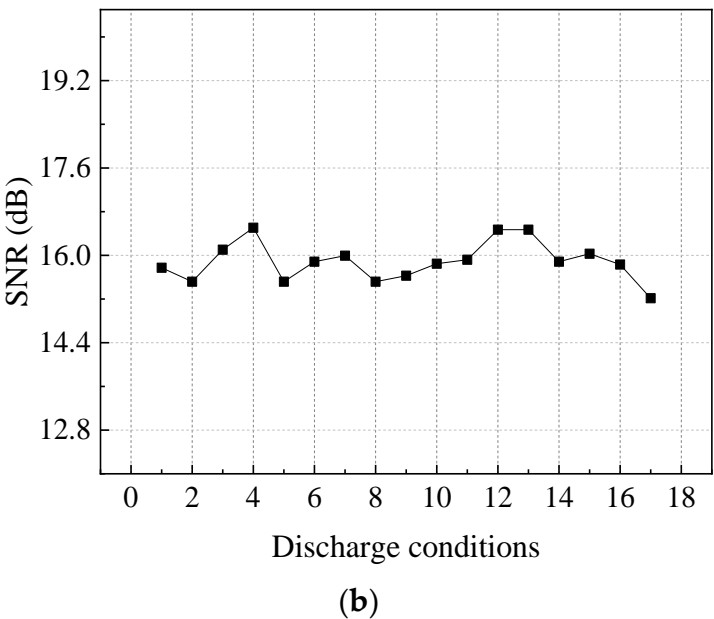

**(b)**

**Figure 16.** RMS of displacement before and after filtering (**a**) and its SNR (**b**) of signal at T9.

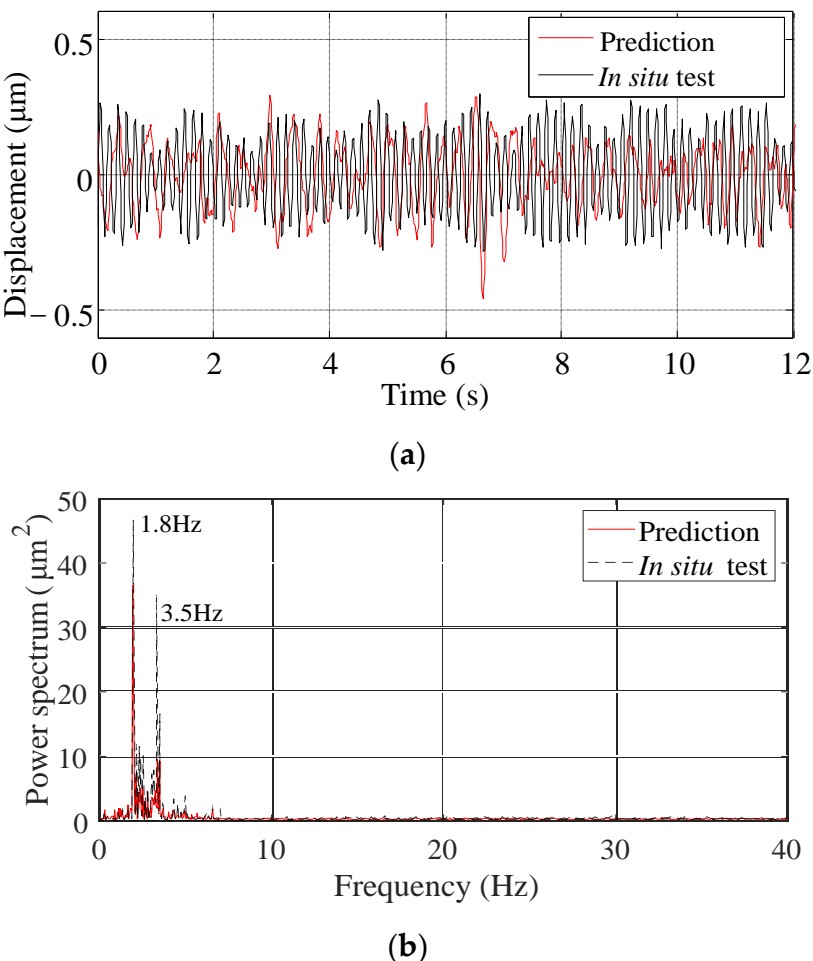

**(a)**

**(b)**

**Figure 17.** The signal of the prediction and the in situ measurement of signal at T9: (**a**) Time history curves; (**b**) CS.

Then, steps (3)–(4) from Section 3.5 are repeated to predict the RMS of the displacements at T9 under 46 discharge conditions happened during the flood season in the year 2013 (July 2013 to October 2013) of Xiangjiaba Dam. The MP-FRF is calculated with the input (displacement at the orifice) and the output (displacement at T9) under 12 similar flood discharge conditions (Figure 8). The noise signal is calculated to be 15.90 dB using Equation (11). The prediction results are compared with the in-situ measurements shown in Figure 18. In terms of the vibration amplitude, the prediction result shows good agreement with the in-situ ones. The error of the overall amplitude is 9.26%. The same increasing trend, reflecting the effects of higher flow rate, and the similar drastically fluctuating patterns, reflecting the effects of different discharge modes, indicate that the predicted results obtained using the MP-FRF are sensitive to the factors that affect the ground vibration intensity. The comparison between the prediction result and the in-situ measurement at different distances from the dam area is shown in Table 3. They are T4 and T5 at 0.5 km, T9 and T12 at 1 km, T15 and T18 at 1.5 km, and T19 and T21 at 2 km, respectively. It can be seen that all the prediction errors are less than 10%, and the error increases with the increase of distance.

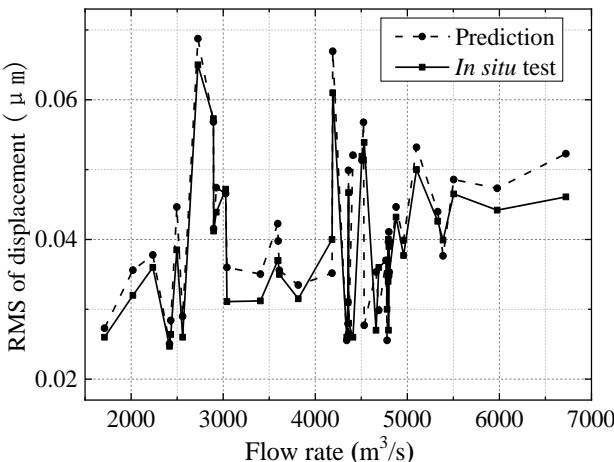

**Figure 18.** Comparison between prediction and in situ measurement RMS of displacement at T9.

**Table 3.** The comparison between the prediction result and the in situ measurement at different distances from the dam area.

| Measurement Points | T4 | | T5 | | T9 | | T12 | | T15 | | T18 | | T19 | | T21 | |
|---|---|---|---|---|---|---|---|---|---|---|---|---|---|---|---|---|
| Flow Rate (m3/s) | I (μm) | P (μm) | I (μm) | P (μm) | I (μm) | P (μm) | I (μm) | P (μm) | I (μm) | P (μm) | I (μm) | P (μm) | I (μm) | P (μm) | I (μm) | P (μm) |
| 1711 | 0.010 | 0.011 | 0.008 | 0.009 | 0.026 | 0.027 | 0.008 | 0.008 | 0.018 | 0.019 | 0.008 | 0.008 | 0.005 | 0.007 | 0.007 | 0.007 |
| 2020 | 0.014 | 0.014 | 0.012 | 0.011 | 0.032 | 0.036 | 0.010 | 0.011 | 0.022 | 0.024 | 0.010 | 0.013 | 0.007 | 0.009 | 0.008 | 0.009 |
| 2236 | 0.015 | 0.017 | 0.012 | 0.012 | 0.036 | 0.038 | 0.011 | 0.012 | 0.028 | 0.031 | 0.013 | 0.012 | 0.010 | 0.011 | 0.009 | 0.009 |
| 2417 | 0.010 | 0.013 | 0.008 | 0.013 | 0.025 | 0.025 | 0.010 | 0.008 | 0.017 | 0.017 | 0.007 | 0.007 | 0.008 | 0.011 | 0.006 | 0.006 |
| 2431 | 0.018 | 0.014 | 0.012 | 0.009 | 0.026 | 0.028 | 0.008 | 0.010 | 0.022 | 0.019 | 0.008 | 0.011 | 0.009 | 0.009 | 0.007 | 0.007 |
| 2497 | 0.016 | 0.018 | 0.012 | 0.014 | 0.039 | 0.045 | 0.012 | 0.014 | 0.026 | 0.031 | 0.014 | 0.013 | 0.012 | 0.012 | 0.010 | 0.011 |
| 2560 | 0.013 | 0.012 | 0.014 | 0.017 | 0.026 | 0.029 | 0.008 | 0.009 | 0.018 | 0.020 | 0.008 | 0.009 | 0.008 | 0.009 | 0.007 | 0.007 |
| 2726 | 0.024 | 0.028 | 0.021 | 0.022 | 0.065 | 0.069 | 0.020 | 0.021 | 0.045 | 0.047 | 0.019 | 0.021 | 0.018 | 0.022 | 0.016 | 0.017 |
| 2894 | 0.027 | 0.023 | 0.018 | 0.018 | 0.057 | 0.057 | 0.020 | 0.017 | 0.041 | 0.039 | 0.012 | 0.016 | 0.020 | 0.018 | 0.014 | 0.016 |
| 2896 | 0.017 | 0.021 | 0.013 | 0.016 | 0.041 | 0.042 | 0.013 | 0.013 | 0.028 | 0.028 | 0.012 | 0.012 | 0.013 | 0.013 | 0.010 | 0.010 |
| 2925 | 0.018 | 0.019 | 0.018 | 0.015 | 0.044 | 0.047 | 0.012 | 0.015 | 0.030 | 0.034 | 0.013 | 0.014 | 0.014 | 0.015 | 0.011 | 0.012 |
| 3027 | 0.019 | 0.019 | 0.015 | 0.015 | 0.047 | 0.047 | 0.014 | 0.014 | 0.037 | 0.032 | 0.012 | 0.015 | 0.015 | 0.015 | 0.012 | 0.012 |
| 3041 | 0.017 | 0.015 | 0.010 | 0.012 | 0.031 | 0.036 | 0.010 | 0.011 | 0.026 | 0.025 | 0.009 | 0.011 | 0.009 | 0.012 | 0.008 | 0.009 |
| 3404 | 0.013 | 0.017 | 0.013 | 0.011 | 0.031 | 0.035 | 0.013 | 0.011 | 0.021 | 0.026 | 0.013 | 0.010 | 0.010 | 0.011 | 0.008 | 0.009 |
| 3593 | 0.018 | 0.020 | 0.012 | 0.014 | 0.037 | 0.042 | 0.011 | 0.013 | 0.028 | 0.029 | 0.011 | 0.013 | 0.012 | 0.014 | 0.009 | 0.011 |
| 3597 | 0.015 | 0.016 | 0.012 | 0.013 | 0.037 | 0.040 | 0.014 | 0.012 | 0.025 | 0.027 | 0.011 | 0.012 | 0.015 | 0.017 | 0.009 | 0.012 |
| 3610 | 0.014 | 0.017 | 0.015 | 0.014 | 0.035 | 0.036 | 0.011 | 0.014 | 0.028 | 0.033 | 0.013 | 0.011 | 0.011 | 0.011 | 0.009 | 0.009 |

**Table 3.** *Cont.*

| Measurement Points | T4 | | T5 | | T9 | | T12 | | T15 | | T18 | | T19 | | T21 | |
|---|---|---|---|---|---|---|---|---|---|---|---|---|---|---|---|---|
| Flow Rate (m3/s) | I (μm) | P (μm) | I (μm) | P (μm) | I (μm) | P (μm) | I (μm) | P (μm) | I (μm) | P (μm) | I (μm) | P (μm) | I (μm) | P (μm) | I (μm) | P (μm) |
| 3816 | 0.011 | 0.014 | 0.010 | 0.011 | 0.032 | 0.033 | 0.010 | 0.010 | 0.022 | 0.023 | 0.009 | 0.013 | 0.010 | 0.011 | 0.008 | 0.008 |
| 4182 | 0.016 | 0.019 | 0.013 | 0.011 | 0.040 | 0.035 | 0.012 | 0.011 | 0.032 | 0.024 | 0.012 | 0.010 | 0.017 | 0.014 | 0.010 | 0.009 |
| 4190 | 0.025 | 0.027 | 0.020 | 0.022 | 0.061 | 0.067 | 0.019 | 0.021 | 0.042 | 0.046 | 0.018 | 0.020 | 0.015 | 0.019 | 0.015 | 0.017 |
| 4342 | 0.016 | 0.010 | 0.016 | 0.013 | 0.026 | 0.026 | 0.015 | 0.008 | 0.018 | 0.018 | 0.013 | 0.008 | 0.008 | 0.012 | 0.007 | 0.009 |
| 4355 | 0.013 | 0.013 | 0.010 | 0.010 | 0.031 | 0.031 | 0.010 | 0.013 | 0.021 | 0.025 | 0.009 | 0.012 | 0.013 | 0.010 | 0.008 | 0.008 |
| 4356 | 0.013 | 0.011 | 0.014 | 0.016 | 0.031 | 0.028 | 0.013 | 0.009 | 0.025 | 0.022 | 0.009 | 0.008 | 0.010 | 0.012 | 0.008 | 0.007 |
| 4361 | 0.019 | 0.020 | 0.015 | 0.016 | 0.047 | 0.050 | 0.014 | 0.015 | 0.032 | 0.034 | 0.014 | 0.015 | 0.015 | 0.016 | 0.012 | 0.012 |
| 4363 | 0.014 | 0.011 | 0.009 | 0.009 | 0.028 | 0.027 | 0.009 | 0.008 | 0.019 | 0.018 | 0.012 | 0.008 | 0.009 | 0.009 | 0.007 | 0.007 |
| 4410 | 0.010 | 0.017 | 0.012 | 0.017 | 0.026 | 0.052 | 0.010 | 0.016 | 0.028 | 0.036 | 0.008 | 0.012 | 0.008 | 0.014 | 0.007 | 0.013 |
| 4507 | 0.017 | 0.021 | 0.017 | 0.017 | 0.052 | 0.051 | 0.016 | 0.018 | 0.036 | 0.035 | 0.015 | 0.015 | 0.017 | 0.017 | 0.013 | 0.013 |
| 4524 | 0.019 | 0.023 | 0.015 | 0.018 | 0.051 | 0.057 | 0.016 | 0.017 | 0.035 | 0.039 | 0.012 | 0.017 | 0.014 | 0.018 | 0.013 | 0.014 |
| 4532 | 0.022 | 0.028 | 0.017 | 0.019 | 0.054 | 0.028 | 0.017 | 0.019 | 0.037 | 0.029 | 0.016 | 0.018 | 0.017 | 0.019 | 0.013 | 0.007 |
| 4660 | 0.016 | 0.014 | 0.009 | 0.011 | 0.027 | 0.035 | 0.014 | 0.011 | 0.019 | 0.024 | 0.008 | 0.011 | 0.009 | 0.011 | 0.007 | 0.009 |
| 4690 | 0.015 | 0.012 | 0.012 | 0.010 | 0.036 | 0.030 | 0.011 | 0.009 | 0.025 | 0.026 | 0.011 | 0.009 | 0.012 | 0.010 | 0.009 | 0.007 |
| 4770 | 0.015 | 0.015 | 0.016 | 0.020 | 0.037 | 0.037 | 0.011 | 0.011 | 0.025 | 0.025 | 0.011 | 0.011 | 0.012 | 0.012 | 0.009 | 0.009 |
| 4780 | 0.012 | 0.010 | 0.010 | 0.008 | 0.030 | 0.026 | 0.012 | 0.012 | 0.021 | 0.028 | 0.009 | 0.008 | 0.010 | 0.008 | 0.008 | 0.006 |
| 4792 | 0.014 | 0.014 | 0.011 | 0.013 | 0.034 | 0.035 | 0.010 | 0.011 | 0.023 | 0.024 | 0.012 | 0.015 | 0.013 | 0.011 | 0.009 | 0.009 |
| 4794 | 0.017 | 0.014 | 0.017 | 0.015 | 0.040 | 0.035 | 0.012 | 0.011 | 0.027 | 0.022 | 0.012 | 0.017 | 0.015 | 0.014 | 0.010 | 0.009 |
| 4796 | 0.011 | 0.016 | 0.012 | 0.013 | 0.027 | 0.040 | 0.012 | 0.012 | 0.024 | 0.027 | 0.008 | 0.012 | 0.009 | 0.012 | 0.007 | 0.010 |
| 4800 | 0.016 | 0.017 | 0.013 | 0.013 | 0.039 | 0.041 | 0.012 | 0.013 | 0.027 | 0.028 | 0.012 | 0.012 | 0.013 | 0.013 | 0.010 | 0.010 |
| 4803 | 0.014 | 0.014 | 0.011 | 0.011 | 0.035 | 0.035 | 0.011 | 0.011 | 0.024 | 0.029 | 0.014 | 0.017 | 0.014 | 0.011 | 0.009 | 0.009 |
| 4878 | 0.017 | 0.022 | 0.014 | 0.014 | 0.043 | 0.045 | 0.013 | 0.014 | 0.030 | 0.031 | 0.013 | 0.013 | 0.012 | 0.014 | 0.011 | 0.011 |
| 4960 | 0.018 | 0.016 | 0.016 | 0.013 | 0.038 | 0.040 | 0.012 | 0.012 | 0.026 | 0.027 | 0.011 | 0.013 | 0.012 | 0.013 | 0.009 | 0.010 |
| 5100 | 0.020 | 0.021 | 0.016 | 0.017 | 0.050 | 0.053 | 0.015 | 0.016 | 0.034 | 0.038 | 0.016 | 0.016 | 0.016 | 0.017 | 0.013 | 0.013 |
| 5330 | 0.017 | 0.018 | 0.014 | 0.018 | 0.043 | 0.044 | 0.013 | 0.014 | 0.029 | 0.030 | 0.013 | 0.013 | 0.014 | 0.014 | 0.011 | 0.011 |
| 5388 | 0.014 | 0.015 | 0.013 | 0.012 | 0.040 | 0.038 | 0.012 | 0.012 | 0.029 | 0.026 | 0.016 | 0.018 | 0.013 | 0.012 | 0.010 | 0.009 |
| 5504 | 0.019 | 0.020 | 0.018 | 0.016 | 0.047 | 0.049 | 0.016 | 0.017 | 0.032 | 0.037 | 0.014 | 0.015 | 0.017 | 0.016 | 0.012 | 0.014 |
| 5976 | 0.018 | 0.024 | 0.018 | 0.020 | 0.044 | 0.047 | 0.014 | 0.015 | 0.034 | 0.036 | 0.013 | 0.015 | 0.014 | 0.015 | 0.011 | 0.012 |
| 6722 | 0.023 | 0.021 | 0.015 | 0.017 | 0.046 | 0.052 | 0.016 | 0.018 | 0.038 | 0.043 | 0.017 | 0.019 | 0.017 | 0.017 | 0.012 | 0.015 |
| Prediction error (%) | 7.07 | | 5.67 | | 9.26 | | 3.09 | | 5.36 | | 9.43 | | 9.12 | | 8.08 | |

## 5. Concluding Remarks

This study proposes a prediction method with a modified FRF and applies it to the ground vibrations induced by flood discharge in the downstream area of Xiangjiaba Dam by using in situ measurement data. The MP-FRF is derived by analyzing major factors such as the noise, system nonlinearity, spectral leakages, and signal latency when the FRF is used. The effects of two types of vibration source as input are quantified. The impact of noise on the predicted amplitude is corrected based on the characteristics of the measured signal. The proposed method involves four steps: signal denoising, MP-FRF estimation, vibration prediction, and noise correction. The following results are obtained in this study:

1.  As the MP-FRF is used to predict vibrations with a broadband frequency and two or more frequency bands with relative high energies, the prediction results show some frequencies caused by non-vibration sources, and the vibration amplitude is amplified. Therefore, the input and output signals need to be filtered, and the amplitude prediction loss caused by filtering can be corrected by adding a constructed white noise signal to the prediction result.

2.  Compared with using the signal at multiple vibration sources after superimposed as the input, using the main source (displacement at the orifice) as the input improves the accuracy of the predicted frequency distribution, and the predicted signals have fewer frequency peaks.

3.  The predicted amplitude errors for the downstream area of Xiangjiaba Dam are less than 10%. The predicted results, like the in-situ measurements, are sensitive to factors that affect the ground vibration intensity, such as the flow rate and different discharge modes. The proposed method can predict the dominant frequency and the frequency bands with relative high energies of the downstream ground vibration. The main vibration propagation band is 1.0–10.0 Hz. The MP-FRF remains stable when the vibration source (input) and the vibration response (output) are selected, and the amplitude of the MP-FRF decreases as the distance increases in both the ancient watercourse area and the east town. Therefore, it can be used to predict some other conditions when the inputs are known and the outputs are unknown.

The proposed method can predict ground vibrations in a timely manner when the signal at the vibration source is measured. The ground vibration intensity can be mitigated by controlling or tuning the discharge conditions by, for example, changing the flow rate, changing the opening method of the orifice, and changing the upstream or downstream water level. In addition, for ground vibrations occurring downstream of the Xiangjiaba Dam, the vibration at the orifice is a very strong main vibration source. This may explain why using the main vibration source as the input is better than using multiple vibration sources as the input. In the future, we will conduct further studies in this area and complement the conclusions of the present study.

**Author Contributions:** Conceptualization, Y.Z. (Yan Zhang) and J.L.; Formal analysis, Y.Z. (Yan Zhang) and G.Z.; Funding acquisition, Y.Z. (Yan Zhang) and Y.L.; Investigation, Y.Z. (Yan Zhang) and Y.L.; Methodology, Y.Z. (Yan Zhang) and S.L.; Project administration, J.L.; Resources, S.L.; Writing—original draft, Y.Z. (Yan Zhang) and Y.Z. (Yanbin Zhao); Writing—review & editing, Y.Z. (Yan Zhang), Y.Z. (Yanbin Zhao) and S.L. All authors have read and agreed to the published version of the manuscript.

**Funding:** The author(s) disclosed receipt of the following financial support for the research, authorship, and/or publication of this article. This work was supported by the National Key R&D Program of China (Grant no. 2018YFC0406900, 2018YFC0406700), the National Natural Science Foundation of China (Grant no. 51709280, 51779277), Applied Basic Research Plan of Shanxi Province (201901D211020).

**Institutional Review Board Statement:** Not applicable.

**Informed Consent Statement:** Informed consent was obtained from all subjects involved in the study.

**Data Availability Statement:** The data that support the findings of this study are available from the corresponding author upon reasonable request.

**Conflicts of Interest:** The authors declared no potential conflict of interest with respect to the research, authorship, and/or publication of this article.

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
