# Peer review of "Predicting Dam Flood Discharge Induced Ground Vibration with Modified Frequency Response Function"

_water, doi:10.3390/w13020144_

Round 1

Reviewer 2 Report

The review of:

Predicting Dam Flood Discharge Induced Ground Vibration with Modified Frequency Response Function

Authors: Yan Zhang, Jijian Lian, Yanbing Zhao, Guoxin Zhang, Yi Liu and Songhui Li

Dear Editor,

I kindly appreciate that, to be offered to review the manuscript. Everyday, we are dealing with unwanted vibrations propagating and perception from wide range of sources around us (traffic, machines etc). This manuscript is presenting the case study focused on the improvement in the transfer function determination between the source and receiver for long distances via complex terrain. The goal is rather challenging and after reading the text I have a few comments.

In following you can find my comments:

  • Page 1, line 20: Please remove the dot at the end of the line, the sentence doesn’t end. “…effectively preventing. damage.”
  • Figure 1: Please add the scale into the figure. It will give reader better imagination.
  • In the figure 1 we can see several blue columns indicating the “relative displacement rms”. Please support this data by additional table, representing at least RMS of displacement of each measurement point in absolute amplitudes or levels.
  • What do authors consider as a limiting value of displacement amplitude, that can occur in the city and will not damage buildings/ negatively affect habitants.
  • Page 4, Eq.1 and 2. The equations are well known from basics physics also from another fields. What do authors think about the importance of time domain. The case is dealing with vibration energy propagation on long distances, the effect of signal latency may not be neglected (that can be also discussed with relation to the Coherence). Please extend the manuscript by discussion about time domain influence on the prediction accuracy. How long time signal was used for analysis? The H will be surely different if the analysis is done based on longer signal.
  • Page 4, Line 123. Please add here some reference, where a reader can get more information about the IFT.
  • Page 4, Line 141. In the manuscript, several times is used a phrase “spectrum of random noise”. Would not be better use the term “Background Noise”?
  • Page 4 MF-FRF. The prediction process may be introduced and explained much better. The text in this chapter is a bit confusing and reader can easily “lose thread”. Please, revise the part 3.2.2 and modify it in the such way.
  • Based on the explanation in the text and figure 3, it seem to be a mistake in the Eq. 3 (the minus and plus signs should be interchanged). Please double check the text, figure and equations.
  • Page 7, Figure 5a. The x- axis is uncomplete.
  • Page 7, In figures from 5 to 13 there are missing marking images with characters (a) (b). Please edit all figures.
  • Figures 5b, 6a, 8. Please modify y-axis in plots to logaritmic.
  • Which value of Coherence do authors consider as sufficient? If we would discuss the vibration propagation on shorter distances, the 50% coherence would be way, way bad.
  • Figures 7b, 9b, 10, 13. Please plot the in situ and prediction curve in the same graph. It will be much easier to compare. Also, please use a log scale at the y-axis.
  • I miss it as a tool for comparing prediction and measurement, comparison in the form of a single number indicator e.g. RMS. Please, compare both cases also by using RMS.
  • Page 8; The basic principle is not properly explained in the text, based on what the prediction works. Is it a transfer function measurement and a derivation of the current state of oscillation at the observation site? Please add such explanation to the manuscript. However, if that is true, the method is mainly based on a transfer function determination (measurement), if that is true, I am not convinced whether this should be considered as a method of prediction and therefore I would fundamentally intervene in the text and especially change the title of the article.
  • Figure 5a, 7a, 9a, why did the authors decide to use the designation on the y-axis Amplitude and not Displacement?
  • Figure 10. Images are overlapping. Please modify the Figure.
  • Page 14. Please add to the text also table including the comparison of measurements and prediction results at all measurement points (not just from position 9). That will make relatively objectivised picture about accuracy of the “prediction”.

After reconsidering all my comments, I’d like to ask authors to resubmit the manuscript for second round of assessment.

Thank you for understanding.

Reviewer

Round 2

Reviewer 1 Report

NA

Author Response

The authors would like to thank the Editor and the Reviewers for the positive feedback. The manuscript has been revised accordingly based on the comments of the Editor and the Reviewers. Typos and grammar mistakes are check thoroughly and corrected accordingly. Some expressions in abstract, introduction, methods and results are modified to make the contents clearer. The added/revised text in the revised manuscript is marked with red color for distinction.

Reviewer 2 Report

Thank you for considering my comments. Hereby I recommend the manuscript to be accepted.

Author Response

(The authors gave the same response as above.)
